# In-Situ Formation of Modified Nickel–Zinc-Layered Double Hydroxide Followed by HPLC Determination of Neonicotinoid Insecticide Residues

**DOI:** 10.3390/molecules27010043

**Published:** 2021-12-22

**Authors:** Jitlada Vichapong, Rawikan Kachangoon, Rodjana Burakham, Yanawath Santaladchaiyakit, Supalax Srijaranai

**Affiliations:** 1Creative Chemistry and Innovation Research Unit, Department of Chemistry and Center of Excellence for Innovation in Chemistry, Faculty of Science, Mahasarakham University, Maha Sarakham 44150, Thailand; Rawikan_wivew@hotmail.com; 2Materials Chemistry Research Center, Department of Chemistry and Center of Excellence for Innovation in Chemistry, Faculty of Science, Khon Kaen University, Khon Kaen 40002, Thailand; rodjbu@kku.ac.th (R.B.); supalax@kku.ac.th (S.S.); 3Department of Chemistry, Faculty of Engineering, Rajamangala University of Technology Isan, Khon Kaen Campus, Khon Kaen 40000, Thailand; sanyanawa@gmail.com

**Keywords:** in situ, layered double hydroxide, neonicotinoids, extraction, honey samples

## Abstract

A single-step preconcentration procedure using the in-situ formation of modified nickel–zinc-layered double hydroxides (LDHs) prior to high-performance liquid chromatography (HPLC) is investigated for the determination of neonicotinoid insecticide residues in honey samples. The LDHs could be prepared by the sequential addition of sodium hydroxide, sodium dodecyl sulfate, nickel nitrate 6-hydrate and zinc nitrate 6-hydrate, which were added to the sample solution. The co-precipitate phase and phase separation were obtained by centrifugation, and then the precipitate phase was dissolved in formic acid (concentrate) prior to HPLC analysis. Various analytical parameters affecting extraction efficiency were studied, and the characterization of the LDHs phase was performed using Fourier-transformed infrared spectroscopy and scanning electron microscopy. Under optimum conditions, the limit of detection of the studied neonicotinoids, in real samples, were 30 μg L^−1^, for all analytes, lower than the maximum residue limits established by the European Union (EU). The developed method provided high enrichment, by a factor of 35. The proposed method was utilized to determine the target insecticides in honey samples, and acceptable recoveries were obtained.

## 1. Introduction

Neonicotinoid insecticides are a new group of insecticides functioning as acetylcholine receptor agonists [1]. They are highly efficient against insects and have relatively low toxicity for humans and mammals; therefore, they are frequently used in agriculture [2]. Due to their high mobility, they may be widely distributed in fruits and vegetables, as well as in the aqueous environment, through runoff and reaching from soil into ground and surface waters and in the vicinities of agricultural areas [3,4]. In order to protect consumers, most nations and organizations, such as the European Union (EU), have established stringent maximum residue levels (MRLs) in different matrices. The MRLs of neonicotinoid insecticides are between 0.01 and 1.50 mg kg^−1^ [5]. Therefore, simple and sensitive analytical methods are required for monitoring these compounds in the environment.

Due to their high polarity and low volatility, neonicotinoid insecticides are usually determined by high-performance liquid chromatography (HPLC) with different detection methods, including ultraviolet [6,7], diode arrays [8], fluorescence [9] and mass spectrometry [10]. Although a MS detector provides more sensitivity and selectivity than a UV-based detector for monitoring target compounds in complex samples, it is a very expensive and complex instrument [11]. Neonicotinoids usually occur in environmental matrices at low concentrations in environmental samples, therefore, a simple sample preparation method and a highly sensitive multi-residue determination approach are highly demanded.

Conventional sample preparation methods, such as solid phase extraction [12] and liquid–liquid extraction (LLE) [13] have been used. However, these methods require large amounts of toxic organic solvents and time-consuming. Recently, layered double hydroxides (LDHs) have acquired enormous consideration. LDHs or hydrotalcite-like compounds are, collectively, a group of two-dimensional layered inorganic materials [14] with positively charged outer layers and exchangeable interlayer anions [15]. They can be used for various purposes, such as anion exchangers, drug carriers, polymer additives, catalysts and precursors [16,17]. The general formula of LDHs is [M_1−x_
^2+^M_x_^3+^(OH)_2_]^x+^ [X*_x_*_/*m*_*^m−^*]·*n*H_2_O, where M^2+^ and M^3+^ are divalent and trivalent metal cations that occupy octahedral positions in the brucite-like layer, and X*^m−^* represents the structural balance anions [17,18,19]. In these materials, divalent cations are partially replaced by trivalent cations, thus generating positive charges that are neutralized by anions and water molecules [20]. LDHs are a new group of green alternative adsorbents, employed because of their non-toxicity [19]. It is well known that organic compounds could be interpolated into LDH by using the electrostatic interaction or hydrophobic interaction [15]. In order to change the properties of LDHs from hydrophilic to hydrophobic, anionic intercalation agents (such as sodium dodecyl sulfate (SDS)) have been added to them, improving their efficacy in adsorbing various organic compounds. The steps of their synthesis require special conditions (i.e., high temperature and pressure) and take a long time (more than 10 h). Surfactant-modified LDH coated magnetic nanoparticles have also been investigated for the HPLC analysis of various pollutants, including phthalate esters [21], phenoxy acid herbicides [22] and organophosphorus pesticides [23]. To the best of our knowledge, the application of dissolvable LDHs and an anionic surfactant to the analysis of neonicotinoids has not been reported. Therefore, a single-step preconcentration, without the synthesis of sorbents, using modified LDHs by means of an anionic surfactant to improve the extraction efficiency of non-ionic analytes, is an attractive goal.

The aim of this study was to investigate the in-situ formation of modified nickel–zinc layered double hydroxide, in a method coupled to a HPLC-UV detector for analysis of neonicotinoid insecticides in honey samples. Ni–Zn-layered hydroxide salts (NZL) were selected as an alternative adsorbent in this study. SDS was used to increase the extraction efficiency of LDHs from non-ionic analytes. Several experimental factors influencing the extraction efficiency were investigated. The optimized method was successfully applied in the analysis of neonicotinoids in honey samples.

## 2. Results and Discussion

### 2.1. Optimization of the In-Situ Formation Procedure of Modified Ni–Zn-Layered Double Hydroxide 

Various parameters influencing the extraction efficiency of the neonicotinoids were optimized, including the concentrations of Zn^2+^, Ni^2+^, SDS and NaOH. In this work, the optimization of each parameter was studied using the one-parameter-at-a-time method. The efficiency of the procedure for the in-situ formation of layer-modified double hydroxide was evaluated in terms of peak area of neonicotinoids.

In this work, Ni–Zn (NO_3_) LDH was selected as the nanostructure for the extraction of neonicotinoid insecticides. The concentrations of Zn^2+^ were studied in the range of 0.005–0.05 mol L^−1^; the results are shown in Figure 1a. It was found that a high extraction efficiency, in terms of peak area, was obtained at the Zn^2+^ concentration of 0.015 mol L^−1^. As a result, Zn^2+^ at 0.015 mol L^−1^ was selected for further study. Additionally, the concentrations of Ni^2+^ were also studied in the range of 0.005–0.05 mol L^−1^. As can be seen in Figure 1b, the extraction efficiency of the studied compounds increased with increasing concentration of Ni^2+^, up to 0.035 mol L^−1^, and then remained constant. Therefore, the Ni^2+^ concentration of 0.035 mol L^−1^ was used. 

LDH was prepared by the co-precipitation method under basic conditions (pH > 8) in order to form a metal hydroxide adsorbent [19]. The effect of NaOH concentration was studied in the range of 0.005–0.05 mol L^−1^ and the results are shown in Figure 1c. It was found that the highest extraction efficiency was observed when the concentration of NaOH was 0.025 mol L^−1^, beyond which increasing NaOH concentrations resulted in low extraction efficiency due to the complete constitution of metal hydroxides in the solution. In addition to the complete formation of metal hydroxides, this may have been due to competition between excess anions (e.g., OH ^–^, Cl^–^, and the anionic surfactant) in the solution. Consequently, NaOH at a concentration of 0.025 mol L^−1^ was used.

In order to improve the extraction efficiency of hydrophobic compounds, the interlayer surfaces of LDHs should be modified with organic anionic surfactants (e.g., SDS) to convert the hydrophilic surface of LDHs to a hydrophobic surface [19], which results in an improvement of the extraction efficiency for hydrophobic compounds. In the present work, SDS was used. The concentration of SDS were studied (0.0025, 0.0069, 0.0115, 0.0161 and 0.025 mol L^−1^), as shown in Figure 1d. The extraction efficiency for the target analytes increased with increasing concentrations of SDS, because SDS strongly interacts with the positively charged LDHs (i.e., strong electrostatic attraction) and the target insecticides (i.e., hydrophobic interaction and van der Waals forces) [21]. It was found that the highest peak area was observed when SDS centration was 0.0115 mol L^−1^.

Before being subjected to HPLC analysis, the LDHs phase should be dissolved, to decrease viscosity, and, specifically, can be dissolved in an acidic medium (pH less than 4) [24]. Two kinds of acid (acetic acid and formic acid) were studied, at volumes of 150 μL, for dissolving the phase; both could dissolve the precipitate phase. However, when using acetic acid, the phase re-precipitated into the solution. Therefore, formic acid was selected as the solvent. Volumes of formic acid (concentrate) were studied in the range of 50–500 μL (data not shown). It was found that a large volume of acid provided poor extraction efficiency due to dilution. Therefore, a volume of formic acid of 150 μL was selected.

Based on the previous studies, the following optimal extraction conditions were used in this method: Zn^2+^ of 0.015 mol L^−1^, Ni^2+^ of 0.035 mol L^−1^, NaOH of 0.025 mol L^−1^, SDS 0.0115 mol L^−1^, vortexed at 1500 rpm for 30 s and centrifuged at 3000 rpm for 5 min.

### 2.2. Characterization of the LDHs Phase Using Fourier-Transformed Infrared Spectra (FTIR) and Scanning Electron Microscope (SEM) Analysis

Fourier-transformed infrared spectroscopy (FTIR) is a useful tool for the characterization of LDHs, involving vibrations in the octahedral lattice, the hydroxyl groups and the interlayer anions [25]. The absorption band around 3420 cm^−1^, shown in the FTIR spectrum of the Ni–Zn (NO_3_^−^) LDH precursor (Figure 2), can be assigned to the stretching vibration of the hydroxyl groups (-OH) of the LDH layers and the interlayer water molecules. The bending mode of water molecules is responsible for the weak band at 1639 cm^−1^ and the strong bands at 1490 and 1383 cm^−1^ are due to the presence of the nitro compound and nitrate ions [26], indicating an incorporation of NO_3_^−^ anions at the interlayer region. In LDH–SDS blank and LDH–SDS standard, absorbance bands appear at 3420 cm^−1^, corresponding to the O–H bonds in the LDH layers and the interlayer water molecules, while methyl (–CH_3_) and methylene (–CH_2_–) appear at approximately 2921 and 2852 cm^−1^. Other peaks, at 1639, 1490 and 1383 cm^−1^, correspond to bending mode of water molecules, the nitro compound and nitrate ions, respectively. Moreover, the region between 1300 and 700 cm^−1^ is probably relative to Metal-O stretching on the structure layer [27]. It is confirmed that Ni(NO_3_)_2-_and-Zn(NO_3_)_2_-layered double hydroxides can be generated under the selected conditions.

SEM images of SDS–LDHs (Ni–Zn hydroxide) and SDS–LDHs (Ni–Zn hydroxide) after the extraction of the target neonicotinoids are shown in Figure 3a,b, respectively. Regarding the arrangement of Ni–Zn, this material is mainly asymmetrical and crystalline. At larger magnifications, in Figure 3b, it is also apparent that the nanosorbent is composed of a pack of several uneven, irregular and polygonal particles. This structure contains abundant active sites and has a high surface area. Therefore, this porous and uniform surface plays a vital role in increasing the interaction between the sorbent and analytes and enhancing the extraction efficiency [28]. The morphology of LDH (Ni–Zn) leads to a loss of octahedral coordination by the opening of one side of the asymmetric crystal on the interlamellar domain, which causes an additional coordinate with one interlamellar water molecule [29]. When SDS is applied to the extraction of the studied neonicotinoids ower-like aggregates seem to be obtained. This phenomenon may be attributed to the adsorption or penetration of the neonicotinoid insecticides in the LDHs phase (Figure 3b).

### 2.3. Evaluation of the Method’s Performance

Using the optimal extraction conditions, the analytical performance of the proposed method was evaluated in term of linearity, coefficient of the determination (R^2^), limits of detections (LODs), limits of quantification (LOQ), precision and enrichment factor (EF). The analytical performances of the proposed method are shown in Table 1. Linearity was in the range of 15–1000 µg L^−1^, with an R^2^ higher than 0.99. LOD and LOQ were defined as the concentration of the target analytes giving a signal-to-noise ratio of 3 (S/N = 3) and 10 (S/N = 10), respectively. The obtained LOD was 0.005 µg L^−1^, while the LOQ was 0.015 µg L^−1^ for all analytes. Precision, in terms of intra-day (*n* = 9) and inter-day (*n* = 9 × 3 days), was investigated as the RSDs of retention time (t_R_) and peak area of the studied compounds. Good precisions, with RSDs less than 2.27 and 4.07% for retention time (t*_R_*) and peak area, respectively, were obtained. The EF, defined as the concentration ratio of the analytes in the settled phase (*C_set_*) and in the aqueous sample (*C_o_*), were 35 for all target analytes. Figure 4 shows chromatograms of standard neonicotinoids obtained by (a) the standard without preconcentration, (b) the standard with preconcentration; the concentration of all standards was 100 µg L^−1^. These results show that the developed extraction method, coupled to HPLC, can increase the sensitivity of detection.

### 2.4. Application to Honey Samples

The applicability of the recommended microextraction method was evaluated in five honey samples. Matrix-match calibration (50–500 μg L^−1^) was used to calculate the quantitation of the neonicotinoids. In the present study, no studied neonicotinoids were detected in the studied samples. The LOD of the neonicotinoids was 30 μg L^−1^ for all analytes. To evaluate the accuracy of the proposed method, the honey samples were spiked with the target insecticides at different concentrations, of 50, 75 and 100 µg L^−1^, before extraction and analysis. The relative recoveries of the studied neonicotinoids are summarized in Table 2. It was found that the relative recoveries of the studied neonicotinoid insecticides were between 81 and 115%, with an RSD of less than 7.6%. Good recoveries were obtained, emphasizing that the proposed method was effective and reliable for the determination of the studied neonicotinoid residues in the honey sample matrices. The chromatograms of the honey and spiked honey samples are shown in Figure 5.

### 2.5. Comparison of the Proposed Method with Other Sample Preparation Methods

The analytical characteristics of the proposed method, in combination with HPLC–UV for the extraction and determination of the target analytes, were compared with those of the other previously published methods. Some analytical parameters of the reported methods and the proposed method itself are summarized in Table 3. In comparison with other methods, the presented method provides lower LODs and high recoveries. According to the results, the proposed method is a rapid, sensitive and repeatable technique that can be used for the preconcentration and determination of target analytes in different samples.

## 3. Materials and Methods

### 3.1. Chemicals and Reagents

All reagents and standards were of at least analytical reagent grade. The neonicotinoid insecticide standards, including acetamiprid, was obtained from Dr. Ehrenstorfer GmbH (Augsburg, Germany), and thiacloprid was obtained from Fluka (Leipzig, Germany. Stock standard solutions of each neonicotinoid insecticides (1000 μg mL^−1^) were prepared in methanol and kept at 4 °C until used. Working standard solutions were prepared by diluting the stock standard solution with water. Deionized water was used throughout the experiments and obtained using a RiOs^TM^ Type I Simplicity 185 (Millipore Waters, Milford, Massachusetts, USA) with a resistivity of 18.2 MΩ.cm. Methanol and acetonitrile of HPLC grade and sodium dodecyl sulfate (SDS) were acquired from Merck (Darmstadt, Germany). Nickel nitrate 6-hydrate and zinc nitrate 6-hydrate were obtained from Elago Enterprlses pty (New South Wales, Australia). Sodium hydroxide (NaOH) was obtained from Ajax Finechem (New South Wales, Australia). Formic acid was provided by JT Baker (Phillipsburg, NJ, United States).

### 3.2. Instrumentations

Chromatographic analysis was performed on a Waters 1525 Binary LC system (Waters, MA, USA). The analytical column was Chromolith^®^ RP-18C (100–4.6 μm) column (Merck, Germany); the mobile phase consisted of an isocratic elution of acetonitrile and 0.1% formic acid, at a flow rate of 1 mL min^−1^ and detection at 254 nm. A Rheodyne injector furnished with a sample loop of 20 µL was used. The Empower 3 software was used in the for control system. Fourier-transformed infrared spectroscopy (FTIR) (Bruker Invenio-S FT-IR (Bruker Corp, MA, USA) and scanning electron microscope (SEM) (Model JEOL JSM-6460LV, Jeol Canada Inc., St-Hubert, QC, Canada) spectra were used for the characterization of the functional groups and the morphology of the LDH phase. Diamond lens attenuated total resistance (ATR) was also used.

### 3.3. In-Situ Formation of Modified Ni–Zn-Layered Double Hydroxide Procedure

The sample or standard solution (10.00 mL) was mixed with 0.025 mol L^−1^ NaOH, 0.010 mol L^−1^ SDS, 0.035 mol L^−1^ Ni(NO_3_)_2_ and 0.015 mol L^−1^ Zn(NO_3_)_2_ in a 15-mL screw cap centrifuge tube. After that, the tube was vortexed at 1500 rpm for 30 s and then centrifuged at 3000 rpm for 5 min to achieve phase separation. The upper aqueous phase was removed with a syringe. Then, the sediment phase was dissolved with 150 μL formic acid to reduce viscosity. Finally, 20 µL of the mixture was directly injected into the HPLC system for analysis. Figure 6 shows the schematic diagram the proposed in-situ solid phase formation of modified Ni–Zn-layered double hydroxide method.

### 3.4. Preparation of Honey Samples

Five different brands of honey samples were obtained from a supermarket in Maha Sarakham, northeastern Thailand. Honey samples (1.0 g) were weighted and then passed through 10-mL volumetric flasks and diluted with water to the marker. Then it was filtered through a Whatman (no. 42) filter paper to get rid of particulate matter. After that, it was passed through a 0.45-mm nylon membrane filter before extraction. For the accuracy evaluation, the samples were spiked with standard neonicotinoids at different concentrations into the homogenized samples prior to the extraction.

### 3.5. Method Validation Studies

Linearity was evaluated by constructing matrix-matched working calibration curves. Matrix-matched working calibration standard solutions were prepared by spiking appropriate volumes of the standard working solution to 1.0 g of the blank sample. After that, the proposed microextraction procedure was conducted, as previously described. The calibration standards were run in triplicate and the average correlation coefficient values are reported. A series of matrix-matched calibration standards were prepared in blank sample extracts, and solvent calibration standards also were prepared in pure solvent. Precision (intra-assay and inter-assay) was obtained by analyzing nine replicates of the spiked sample. Intra-day experiments were performed in one day, and inter-day experiments were performed over three days. Three standard spiked samples at different concentration levels (*n* = 3) were selected for the calculations of the recovery experiments, which were analyzed by comparing the measured concentration data with those of the spiked samples. The relative standard deviation (RSD) of all recovery data was calculated as well, to validate the feasibility of this method. The limits of detection (LOD) and quantification (LOQ) were obtained from the signal-to-noise ratios, 3:1 and 10:1, respectively.

## 4. Conclusions

A procedure for the in-situ formation of modified Ni–Zn-layered double hydroxide has been successfully applied for one-step extraction and preconcentration of neonicotinoid pesticides in honey samples prior to HPLC analysis. In addition, the preparation of Ni–Zn-LDHs without thermally synthesized steps and the addition of an anionic surfactant provided superiority to the typical extraction by LDHs, owing to its ability to extract non-ionic compounds. This extraction method could improve detection sensitivity significantly and it is simple and rapid, without the time- and energy- consuming synthesis step of sorbents. The method provided high sensitivity with low LODs (less than MRLs set by the EU in honey (50–200 μg kg^−1^)). The method could also be useful for the analysis and monitoring of neonicotinoid pesticide residues in a honey matrices. This microextraction method can be used as an alternative method to other complicated, synthesis-based extraction methods.

## Figures and Tables

**Figure 1 molecules-27-00043-f001:**
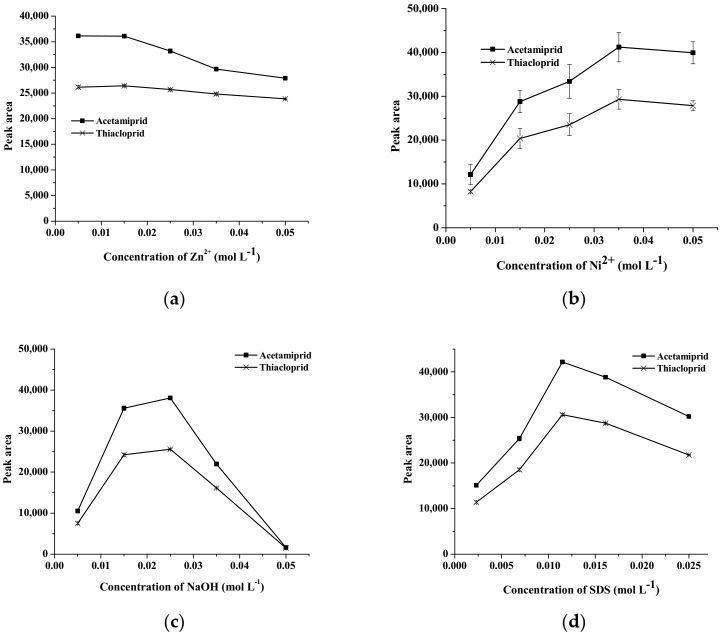
Effect of (**a**) the concentration of Zn^2+^, (**b**) the concentration of Ni^2+^, (**c**) the concentration of NaOH and (**d**) the concentration of SDS.

**Figure 2 molecules-27-00043-f002:**
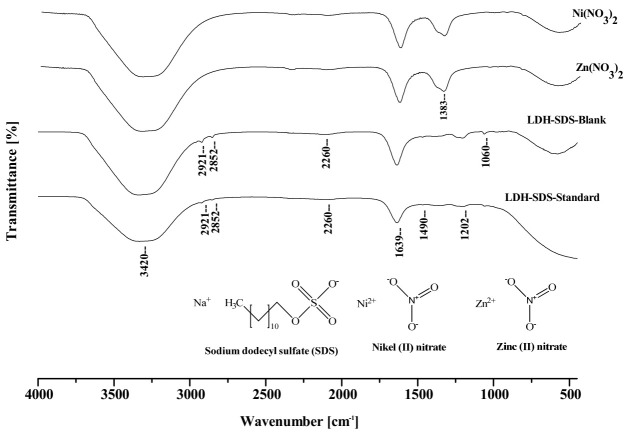
FTIR spectra of Ni(NO_3_)_2_, Zn(NO_3_)_2_ and LDH–SDS–, and LDH–SDS standard after the extraction of the studied neonicotinoids (100 μg L^−1^ each).

**Figure 3 molecules-27-00043-f003:**
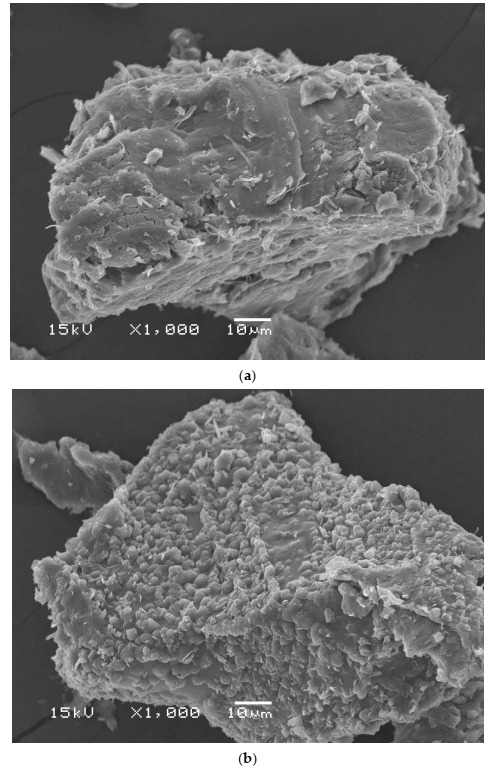
SEM images of (**a**) SDS–LDHs (Ni–Zn hydroxide) and (**b**) SDS–LDHs (Ni–Zn hydroxide) after the extraction of the target neonicotinoids.

**Figure 4 molecules-27-00043-f004:**
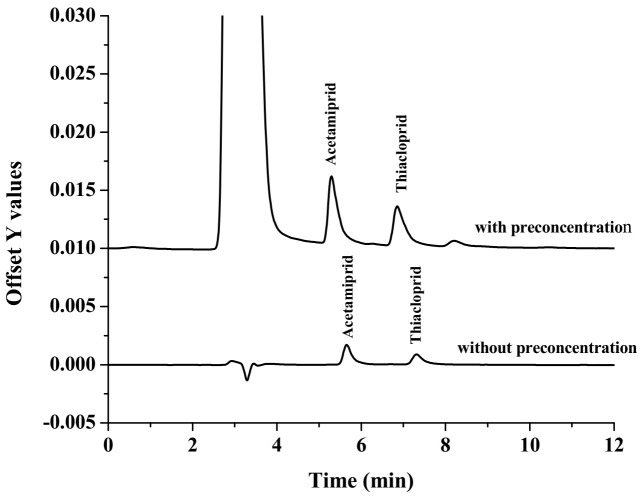
Overlaid chromatogram of studied neonicotinoids with and without preconcentration using the proposed in-situ formation of modified Ni–Zn-layered double hydroxide procedure.

**Figure 5 molecules-27-00043-f005:**
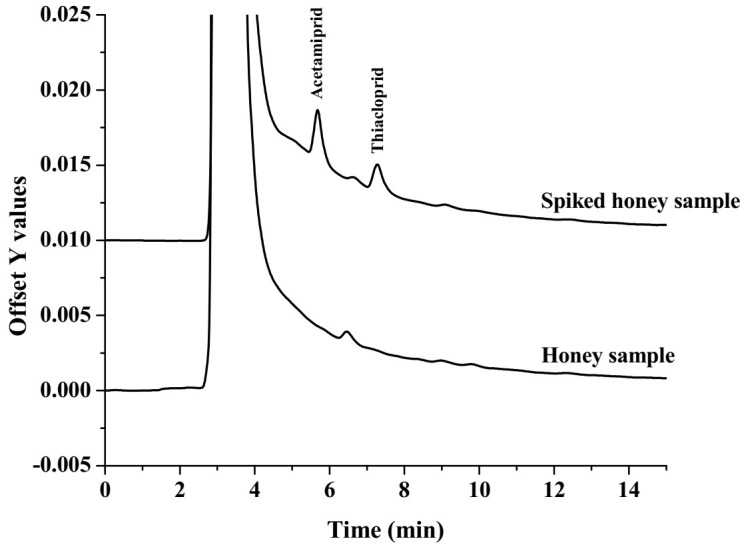
The chromatograms of a honey and spiked honey sample at 50 µg L^−1^.

**Figure 6 molecules-27-00043-f006:**
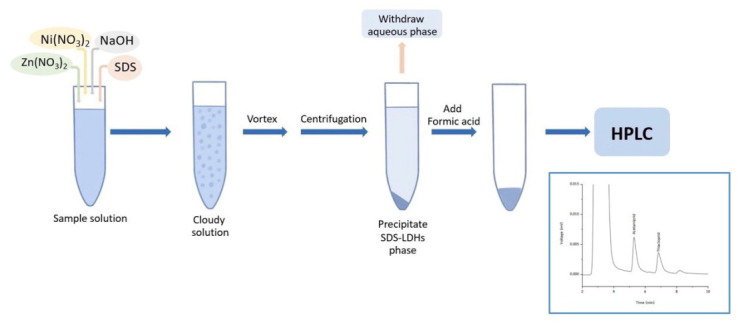
Schematic demonstration of the proposed in-situ solid phase formation of modified Ni–Zn-layered double hydroxide method.

**Table 1 molecules-27-00043-t001:** Analytical performances of the studied neonicotinoids.

Analyte	Linearity (μg L^−1^)	R^2^	LOD (μg L^−1^)	LOQ (μg L^−1^)	Intra-Day Precision (*n* = 9), %RSD		Inter-Day Precision (*n* = 9 × 3), %RSD		EF
					*t* _R_	Peak Area	*t* _R_	Peak Area	
Acetamiprid	15–1000	0.9981	0.005	0.015	1.34	3.73	2.27	4.07	35
Thiacloprid	15–1000	0.9979	0.005	0.015	1.35	1.87	2.05	3.42	35

**Table 2 molecules-27-00043-t002:** Recoveries of the studied neonicotinoids in the spiked samples.

Sample	Spiked (µg L^−1^)	% Recoveries at Different Spiked Levels (% RSD, *n* = 3)	
		Acetamiprid	Thicloprid
Honey I	50	86 (3.2)	115 (7.6)
	75	88 (1.4)	111 (5.9)
	100	92 (2.7)	97 (4.5)
Honey II	50	87 (3.6)	98 (2.8)
	75	92 (7.6)	87 (4.1)
	100	98 (2.1)	89 (4.6)
Honey III	50	98 (2.5)	94 (2.6)
	75	95 (2.6)	81 (4.3)
	100	90 (1.8)	93 (4.2)
Honey IV	50	89 (5.7)	88 (5.3)
	75	97 (3.8)	94 (1.3)
	100	102 (3.8)	94 (3.4)
Honey V	50	83 (2.9)	91 (4.8)
	75	85 (1.8)	97 (6.7)
	100	91 (3.6)	91 (3.3)

**Table 3 molecules-27-00043-t003:** Comparison of the proposed method with other methods in the determination of the selected analytes.

Method	Samples	LOD	Recovery (%)	EF	References
DSPE–DLLME	grain	0.002–0.005 mg kg^−1^	76–123	-	[30]
DLLME	cucumber	0.8–1.2 ng g-1	79.7–98	-	[9]
QuEChERS-DLLME	grains (rice, millet, and maize)	0.04–40 μg kg^−1^	62–118	-	[31]
solid-phase extraction	honey and royal jelly	0.25–5.0 µg kg^−1^	72.8–106.5	-	[32]
in-situ formation of modified Ni–Zn-layered double hydroxide procedure	honey	0.005 μg L^−1^	81–115	35	presented method

## Data Availability

The data presented in this study are available on request from the corresponding author.

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
