# Peer review of "In-Situ Formation of Modified Nickel–Zinc-Layered Double Hydroxide Followed by HPLC Determination of Neonicotinoid Insecticide Residues"

_molecules, 2021, doi:10.3390/molecules27010043_

Round 1

Reviewer 1 Report

  1. Introduction section, progress of neonicotinoid analysis in honey matrix should be added.
  2. Line 90, why did authors choose this LDH system?
  3. Line 123, Figure 1-3, should be closed to the related text.
  4. Line 139, “another”.
  5. Line 146, please explain “empty and standard LDH”.
  6. Line 157, Line 163, sentences are confused and should be re-written, e.g., “two Ni-Zn hydroxide two layers” , “The morphological of LDH (Ni-Zn).”
  7. Line 159, where is the Figure 7b?
  8. Line 181, “(“.
  9. Line 185, authors should carefully investigate and state LODs and LOQs. These values are inconsistent with that in Line 207.
  10. Line 216, section 2.5 should be re-organized. Comparison with published methods in honey matrix would be better.
  11. Line 228, section Materials and Methods, details of method validation should be provided. How did authors calculate the concentration of analytes in the settled phase?
  12. Language should be checked.
  13. Errors in references section should be corrected, e.g., ref.12-13, 14-15, 35-36.

Author Response

 We thank you and the reviewers very much for the valuable comments. The manuscript has been carefully revised accordingly to the comments as in the following: Manuscript entitled “In-situ formation of modified nickel/zinc layered double    hydroxide followed by HPLC determination of neonicotinoid insecticide residues

All changes are highlighted in the revised manuscript using red colored.

Thank you very much for consideration of our manuscript.

Yours sincerely,

Authors

Reviewer 2 Report

Manuscript ID: 1460312

 In-situ formation of modified nickel/zinc layered double hy-2 droxide followed by HPLC determination of neonicotinoid in-3 secticide residues

Critical review:

  1. Abstract

Abstract looks like a methodology. There is a need for improvement.

  1. Introduction

<<<<<Conventional sample preparation method solid phase extraction [12] and liquid-liquid extraction (LLE) [13] have been used. But, these methods have limitations such as require large amount of toxic organic solvents and sample solution, and take a lot of time.>>>>>

            This, it seems to me, contradicts the point of doing research.

  1. All paragraphs 48 to 73 are completely incomprehensible. Reading this fragment, one has the impression that many threads have been detached from different contexts. There is a need for improvement.
  2.  

Figures 1-4 in their current form are unacceptable. It would be better to present them in one graph. Unless the authors wanted to expand the article to be more abundant.

  1. Figure 7. What is this phenomenon? What was a reason to present a chromatogram. What does it bring to work?
  2. Tables are just as bloated as Figures 1-4. All the information contained therein could be described in a few sentences in the text.
  3. Methodology

After the research methodology presented in this way, it will be difficult for other scientists to repeat the research.

  1. Conclusions are not a description, but a specific finding. Unacceptable.

Author Response

(The authors gave the same response as above.)

Reviewer 3 Report

The manuscript describes a new analytical method for the determination of neonicotinoid insecticide in honey. I think that the manuscript can be accepted for publication after the application of minor language revisions.

Below are some minor revisions:

  • Line 20: “Various analytical parameters” instead of “Various analytical parameter”
  • Line 23: 30 μg/L seems somehow high in my eyes given that LC-MS/MS can easily reach limits of detections of the order of few ng/L. Is the method fit for purpose?
  • Line 31: “They are highly efficient” instead of “They are high efficiencies”
  • Line 32: “they are frequently used” instead of “ their frequently used”
  • Line 45: “occur” instead of “exhibit”
  • Line 50: Eliminate “are”
  • Line 55: “They can be used” instead of “They can use”
  • Line 70: “application” instead of “ap-plication”
  • Line 127: precipitate phase is a new term for me. Please verify that the definition exists
  • Line 153: Never start a sentence with “Which”. Please rephrase
  • Line 212-213: repetition of word “studied”

Author Response

Dear Editor

We thank you and the reviewers very much for the valuable comments. The manuscript has been carefully revised accordingly to the comments as in the following: Manuscript entitled “In-situ formation of modified nickel/zinc layered double    hydroxide followed by HPLC determination of neonicotinoid insecticide residues

All changes are highlighted in the revised manuscript using red colored.

Thank you very much for consideration of our manuscript.

Yours sincerely,

Authors

Reviewer 4 Report

The research presented by Vichapong et al., describes the use of nickel/zinc layered double hydroxide to determine the concentration of neonicotinoid insecticides residues by HPLC. 

This technology could be a useful tool in the agro alimentary industry since it allows the measurement of residual insecticides.

Some points should be addressed to improve the quality of the manuscript:

Line 55 change carri-ers by carriers

Line 65 change inprove by improve

Line 70 change ap-plication by application

Line 73 change effi-ciency by efficiency

Line 183 and 266 Please remove the extra point

Line 283 – 284 Remove “was added”

Line 316 remove the extra point

In figure 1, homogenize the size of the letters in the graphs

In figure 2, change both Ni++ and Zn++ by Ni2+ and Zn2+

Finally, I strongly suggest editing the manuscript by some English native speaker

Author Response

(The authors gave the same response as above.)

Round 2

Reviewer 1 Report

I recommend acceptance in the present form.

Author Response

(The authors gave the same response as above.)

Reviewer 2 Report

I'm sorry, but I believe the article has not been corrected as recommended. 

Author Response

(The authors gave the same response as above.)
